# Diagnostic Tool for Out-of-Sample Model Evaluation

**Ludvig Hult**                                    *ludvig.hult@it.uu.se*
*Department of Information Technology*
*Uppsala University*

**Dave Zachariah**                                 *dave.zachariah@it.uu.se*
*Department of Information Technology*
*Uppsala University*

**Petre Stoice**                                   *ps@it.uu.se*
*Department of Information Technology*
*Uppsala University*

**Reviewed on OpenReview:** *https://openreview.net/forum?id=Ulf3QZG9DC*

## Abstract

Assessment of model fitness is a key part of machine learning. The standard paradigm of model evaluation is analysis of the average loss over future data. This is often explicit in model fitting, where we select models that minimize the average loss over training data as a surrogate, but comes with limited theoretical guarantees. In this paper, we consider the problem of characterizing a batch of out-of-sample losses of a model using a calibration data set. We provide finite-sample limits on the out-of-sample losses that are statistically valid under quite general conditions and propose a diagonistic tool that is simple to compute and interpret. Several numerical experiments are presented to show how the proposed method quantifies the impact of distribution shifts, aids the analysis of regression, and enables model selection as well as hyperparameter tuning.

## 1 Introduction

Fitting a model to data is a central task in machine learning, signal processing, statistics and other areas (Bishop, 2006; Hastie et al., 2009; Söderström & Stoica, 2001; Kay, 1993; Fitzmaurice et al., 2011). A fitted model $f$ can be assessed by considering a loss function $\ell(\cdot)$ that evaluates the model on future data points. This is called *out-of-sample* analysis, since it considers data points beyond those in the training data sample.

Classical statistical models are often assessed by verifying assumption validity through statistical or graphical tools, a process called model diagnostics, diagnostic checks, or regression diagnostics (Ruppert et al., 2003; Belsley et al., 1980). The term has also been established in a wider sense for general checks to verify a model fitness for given data, see e.g. Casper et al. (2022); Zhang et al. (2022). We propose a diagnostic tool that helps evaluating the model performance on out-of-sample data.

In this paper, we consider the problem of characterizing $m$ out-of-sample losses of a model $f$ using a calibration data set. The choice of $m$ will depend on the application. In online learning, $m = 1$ is reasonable since the model may only be valid for a single prediction. In problems with low data-volumes, such as a predictive health-care model to be used on e.g. 30 future patients with a rare desease, $m = 30$ may be appropriate. For models in high volume applications, such as e-commerce, the limit $m \to \infty$ may be relevant. We derive finite-sample *limits* on the out-of-sample losses that are statistically valid under quite general conditions and for any of the choices of $m$. An illustration is provided in Fig. 1, where we fit a model for predicting house prices to training data and evaluate its absolute prediction error on a calibration data set $\mathcal{D}$. The average

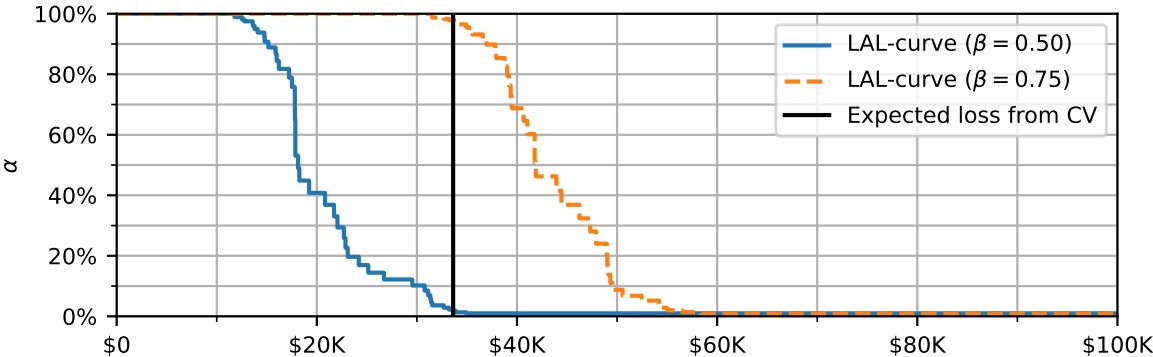

Figure 1: Model error diagonistic. Consider a predictive model $f(X)$ of house price $Y$ in an area described by a feature vector $X$. How will this model perform in $m$ future price predictions? We evaluate the model using the prediction error $\ell(X, Y) = |Y - f(X)|$ as the chosen loss function. The cross-validation method (CV) evaluates the model by estimating the expected out-of-sample loss, which in this case is circa \$36K (indicated by black vertical line). Being an estimate of an expectation, it provides limited information about what individual loss values might be observed. Suppose the model will be used in $m = 100$ prediction instances, and we want to bound at least $\beta = 50\%$ of these future losses. Then the LAL-curve infers that with probability at least $80\%$ (level $\alpha = 20\%$), this fraction of prediction errors will not exceed \$25K (see blue solid curve) For a stronger guarantees, e.g. bounding at least $\beta = 75\%$ of the future losses, the LAL-curve indicates that prediction errors of \$48K must be tolerated at level $\alpha = 20\%$ (see orange dashed curve). Full details for this experiment are provided in Sec. 4.1.

loss on $\mathcal{D}$ is a form of 'cross-validation' and is indicated in the figure. The figure also illustrates the proposed diagnostic statistic: an upper bound $\ell_\alpha^\beta(\mathcal{D})$ on the $\beta$-fraction of $m$ yet unobserved prediction errors that holds with confidence level $1 - \alpha$. By computing this statistical limit $\ell_\alpha^\beta(\mathcal{D})$ as function of $\alpha$, we quantify how probable different out-of-sample losses are for the model $f$. Since this quantification is valid for any size of $\mathcal{D}$, the limit can be employed as a diagnostic tool also in cases where calibration data is scarce or costly to obtain. Moreover, as the validity does not depend on the distribution of the data used to train $f$, the limit can also be used to analyze the severity of distribution shifts, as we will illustrate in the numerical experiments below.

The rest of the paper is organized as follows. We first formalize the general problem of interest, then propose a measure $\ell_\alpha^\beta(\mathcal{D})$ which we refer to as the level-$\alpha$ limit (LAL) on the $\beta$-fraction of $m$ out-of-sample losses and prove its statistical guarantees. This is followed by a series of numerical experiments that demonstrate the utility of LAL. In the closing discussion, the proposed method is related to existing literature.

## 2   Problem Statement

Let $f$ denote a model fitted to data samples $\mathcal{D}_0$ drawn from a distribution $p_0$. We aim to quantify the performance of $f$ on $m$ out-of-sample data points $\{Z_1, \ldots, Z_m\}$ drawn from distribution $p$, which may differ from $p_0$. The performance is quantified using any real-valued loss function $\ell(z)$ the user wants. We let upper case letters denote random variables, e.g., $Z$, and let the lower case version, $z$, represent their realization.

**Example 1** (Density Estimation using a Gaussian model). *A data point is a vector $z \in \mathbb{R}^K$ and we consider a Gaussian density model $f(z) = \mathcal{N}\left(z\,;\,\widehat{\mu},\,\widehat{\Sigma}\right)$, with a fitted mean $\widehat{\mu}$ and covariance matrix $\widehat{\Sigma}$. A common loss function is the negative log-likelihood*

$$\ell(z) = -2\ln f(z) = (z - \widehat{\mu})^{\mathsf{T}}\widehat{\Sigma}^{-1}(z - \widehat{\mu}) + \ln|\widehat{\Sigma}|,$$

*ignoring the constant and scaling by a factor 2.*

**Example 2** (Regression). *A data point is a pair of features $x$ and a label $y$, i.e., $z = (x, y)$. The model $f(x)$ is any estimate of the conditional expectation function $\mathbb{E}[y|x]$. Example models include Gaussian process*

*regressors, random forest regressors, and neural networks. A common loss function is the squared-error loss* $\ell(x, y) = |y - f(x)|^2$.

We assume that we have access to a *calibration data set* $\mathcal{D} = \{Z_1^c, \ldots, Z_n^c\}$ with $n$ samples drawn randomly from $p$, such that the combined data set of $n+m$ samples is exchangeable, i.e., the joint density of the random vector $(Z_1^c, \ldots, Z_n^c, Z_1, \ldots, Z_m)$ is invariant under relabeling of the data points. While the calibration data set $\{Z_1^c, \ldots, Z_n^c\}$ is available to use in computations, the out-of-sample data $\{Z_1, \ldots, Z_m\}$ is future, yet unseen data. Exchangeability includes the common independent and identically distributed (iid) data assumption as a special case. An example of exchangeable *non*-iid data is sampling without replacement from a finite population.

The problem we consider is to characterize $m$ unknown *out-of-sample losses*

$$\{\ell(Z_1), \ldots, \ell(Z_m)\} \tag{1}$$

of the model $f$. Specifically, we want to bound a fraction $\beta \in (0, 1)$ of the $m$ losses with a confidence $1 - \alpha$. That is, we want to find a statistical limit $\ell_\alpha^\beta(\mathcal{D})$ on how large future losses we are likely to observe for the model:

$$\mathbb{P}\Big[\text{at least a fraction } \beta \text{ of losses } \ell(Z_i) \text{ respects } \ell(Z_i) \leq \ell_\alpha^\beta(\mathcal{D})\Big] \geq 1 - \alpha. \tag{2}$$

We will call such a value $\ell_\alpha^\beta(\mathcal{D})$ the *level-$\alpha$ limit* for the $\beta$-fraction of $m$ losses, or a LAL for short.

We call the graph of $\ell_\alpha^\beta(\mathcal{D})$ versus $\alpha$ a *LAL-curve*, as illustrated in, e.g., Fig. 1. By plotting the value $\ell_\alpha^\beta(\mathcal{D})$ on the horizontal axis, we can visualize the tail behaviour of the out-of-sample losses for $f$ in a transparent manner. Thus given a valid LAL, we propose to use the LAL-curve as a diagnostic tool for model evaluation.

## 3 Method

For notational convenience, let $L_i = \ell(Z_i)$ and arrange the $m$ out-of-sample losses (1) in increasing order: $L_{(1)} \leq L_{(2)} \leq \cdots \leq L_{(m)}$. The criterion (2) can be expressed compactly as

$$\boxed{\mathbb{P}\left[L_{(\lceil m\beta \rceil)} > \ell_\alpha^\beta(\mathcal{D})\right] \leq \alpha} \tag{3}$$

and our objective is to find a limit $\ell_\alpha^\beta(\mathcal{D})$ for any given $\beta \in (0, 1]$ and $\alpha \in (0, 1)$.

For the calibration data $\mathcal{D}$, let $L_i^c = \ell(Z_i^c)$ and define the set of losses $\{L_1^c, \ldots, L_n^c\}$ and order statistics $L_{(1)}^c \leq L_{(2)}^c \leq \cdots \leq L_{(n)}^c$. We also define the special cases $L_{(0)}^c$ and $L_{(n+1)}^c$ to be the infimum and supremum of the support of the distribution of $\{L_i^c\}_{i=1}^m$, possibly $\pm\infty$.

### 3.1 General LAL expression

The following result holds generically; it assumes two or more continuously distributed random variables don't take exactly the same value. The probability of such an event is zero. To simplify the reading, we will skip the assertions of 'almost surely' and 'assuming no ties'.

**Theorem 1.** *Let* $\ell_\alpha^\beta(\mathcal{D}) = L_{(k^\star)}^c$, *where*

$$k^\star = \min\left\{k \in \{1, \ldots, n+1\} \mid a(k) \leq \alpha\right\},$$

$$a(k) = \sum_{j=k}^{n+1} \frac{\binom{n-j+m-\lceil m\beta \rceil}{n-j}\binom{j+\lceil m\beta \rceil-1}{j}}{\binom{n+m}{m}} \tag{4}$$

*This is a valid LAL, satisfying* (3).

*Proof.* Define the set of calibration losses $\mathcal{L}^c = \{L_1^c, \ldots, L_n^c\}$ and out-of-sample losses $\mathcal{L} = \{L_1, \ldots, L_m\}$. We will first prove the result in the case of continuous random variables, and then for discrete random variables.

Consider the case when $\mathcal{L}^c \cup \mathcal{L}$ has a continuous joint distribution, so there are no ties. We need to show that

$$\mathbb{P}\left[L_{(\lceil m\beta \rceil)} > L^c_{(k^\star)}\right] \le \alpha. \tag{5}$$

Because of the decomposition into sum

$$\mathbb{P}\left[L_{(\lceil m\beta \rceil)} > L^c_{(k^\star)}\right] = \mathbb{P}\left[L^c_{(k^\star)} \le L_{(\lceil m\beta \rceil)}\right] = \sum_{j=k^\star}^{n} \mathbb{P}\left[L^c_{(j)} \le L_{(\lceil m\beta \rceil)} < L^c_{(j+1)}\right], \tag{6}$$

a closed form expression for $\mathbb{P}\left[L^c_{(j)} \le L_{(i)} < L^c_{(j+1)}\right]$ may lead us to proving (5).

The set of random variables $\mathcal{L}^c \cup \mathcal{L} = \{L^c_1, \dots, L^c_n, L_1, \dots, L_m\}$ can be sorted. Denote these sorted variables $J_{(1)} < J_{(2)} < \dots < J_{(n+m)}$. Every such value $J_{(i)}$ come from an origin set, which is either $\mathcal{L}^c$ or $\mathcal{L}$. This assignment to the origin set partitions the set of $J_{(i)}$-values into two subsets. There are $\binom{n+m}{m}$ such partitions, and they are equally probable, due to the assumed exchangeability.

We next investigate how many of the partitions fulfil $L^c_{(j)} \le L_{(i)} < L^c_{(j+1)}$, for $1 \le i \le m$, $1 \le j \le n$. In this situation, $L_{(i)} = J_{(i+j)}$. Of the $i + j - 1$ losses less than $J_{(i+j)}$, $j$ has $\mathcal{L}^c$ as the origin set. There are thus $\binom{j+i-1}{j}$ equally probable partitions of the lesser loss values. Similarly, the loss values greater than $L_{(i)}$ can be partitioned in $\binom{n-j+m-i}{n-j}$ ways. This shows that

$$\mathbb{P}\left[L^c_{(j)} \le L_{(i)} < L^c_{(j+1)}\right] = \frac{\binom{n-j+m-i}{n-j}\binom{j+i-1}{j}}{\binom{n+m}{m}}. \tag{7}$$

By combining with (6), we have found

$$\mathbb{P}\left[L_{(\lceil m\beta \rceil)} > L^c_{(k)}\right] = \sum_{j=k}^{n} \frac{\binom{n-j+m-\lceil m\beta \rceil}{n-j}\binom{j+\lceil m\beta \rceil-1}{j}}{\binom{n+m}{m}} =: a'(k)$$

Any $k$ such that $a'(k) \le \alpha$ would provide us with a valid LAL. However, since the range of $a'(k)$ is $\left[\frac{1}{\binom{n+m}{m}}, 1\right]$, the case when $\alpha < \frac{1}{\binom{n+m}{m}}$ means no such $k$ exists. Defining $\binom{b}{r} = 0$ for all $b$ and $r < 0$ allows us to finally define $a(k)$ to

$$a(k) := \sum_{j=k}^{n+1} \frac{\binom{n-j+m-\lceil m\beta \rceil}{n-j}\binom{j+\lceil m\beta \rceil-1}{j}}{\binom{n+m}{m}}$$

increasing its domain of definition to $\{1, \dots, n+1\}$ and its range to $[0, 1]$. By selecting $k^\star = \min\{k \in \{1, \dots, n+1\} | a(k) \le \alpha\}$, we have proven the theorem for continuous variables.

Next, we prove the result for discrete random variables. Let $F$ be the joint cdf for $\mathcal{L}^c \cup \mathcal{L}$, where each loss takes values in a finite or countable set $V = \{v_i\}$ of real numbers. Since we may get ties with non-zero probability, the preceding analysis fails. To circumvent this problem, construct a random vector $(\lambda_1, \dots, \lambda_{n+m})$ with cdf $F^*$ such that

$$F^*(l_1, \dots, l_{n+m}) \ge F(l_1, \dots, l_{n+m}) \text{ always}$$
$$F^*(l_1, \dots, l_{n+m}) = F(l_1, \dots, l_{n+m}) \text{ if } (l_1, \dots, l_{n+m}) \in V^{n+m}$$

Define further a set of random variables $\overline{\lambda}_i = \min\{v \in V | v \ge \lambda_i\}$ for all $i$. Now $(\overline{\lambda}_1, \dots, \overline{\lambda}_{n+m})$ is equal to $(L^c_1 \dots L^c_n, L_1, \dots, L_m)$ in distribution. Also, $(\overline{\lambda}_{(i)} > \overline{\lambda}_{(j)}) \Rightarrow (\lambda_{(i)} > \lambda_{(j)})$ for all $i, j$. Together, this means $\mathbb{P}\left[L_{(\lceil \beta m \rceil)} > L^c_{(k^\star)}\right] = \mathbb{P}\left[\overline{\lambda}_{(\lceil \beta m \rceil + n)} > \overline{\lambda}_{(k^\star)}\right] \le \mathbb{P}\left[\lambda_{(\lceil \beta m \rceil + n)} > \lambda_{(k^\star)}\right] = a(k^\star)$. where we have used the result on continuous random variables on $F^*$ to compute $k^\star$. $\square$

**Remark 1.** *The definition of the* LAL, *(2), only demands that the limit level is at least $1 - \alpha$. However, the more conservative $\ell^\beta_\alpha(\mathcal{D})$ is, the larger is the excess coverage. From the proof, we see that if the joint set of losses $(L^c_1 \dots L^c_n, L_1, \dots, L_m)$ has no ties, one may compute the* exact *coverage $1 - a(k^\star)$. Thus when the method is conservative, it is transparently so.*

**Remark 2.** *The proof technique above is inspired by Fligner & Wolfe (1976) which proves the result for the special case of iid data. It is noteworthy that when generalizing from iid to exchangeable data, we keep the same level of precision.*

### 3.2 LAL for a single out-of-sample data point

When $m = 1$, the LAL takes a very simple closed form. By deriving it from basic principles rather than using Thm. 1, we also obtain a tightness guarantee.

**Theorem 2.** *For a single out-of-sample data point (m=1), a LAL can be constructed as $\ell_\alpha^1(\mathcal{D}) = L_{(k^\star)}^c$, where $k^\star = \lceil (n+1)(1-\alpha) \rceil$. For continuous data distributions, the almost sure out-of-sample loss guarantee is*

$$\alpha - \frac{1}{1+n} \leq \mathbb{P}\left[L_1 > \ell_\alpha^1(\mathcal{D})\right] \leq \alpha \tag{8}$$

*For discrete data distributions, only the upper bound in (8) can be guaranteed.*

*Proof.* When $(L_1^c, \ldots, L_n^c, L_1)$ are continuous, the values are almost surely unique, and therefore there are $\binom{n+1}{1} = \frac{1}{n+1}$ equally likely ways to select which one is $L_1$. Only one such selection obeys $L_{(j)}^c \leq L_1 \leq L_{(j+1)}^c$. Therefore,

$$\mathbb{P}[L^c(j) \leq L_1 \leq L_{(j+1)}^c] = \frac{1}{1+n} \text{ and } \mathbb{P}[L_1 \leq L_{(k^\star)}^c] = \frac{k^\star}{1+n}$$

Because $(1-\alpha) \leq \lceil (n+1)(1-\alpha) \rceil/(n+1) \leq (1-\alpha) + 1/(n+1)$ we compute

$$\alpha - \frac{1}{1+n} \leq \mathbb{P}[L_1 > L_{(k^\star)}^c] \leq \alpha$$

When $(L_1^c, \ldots, L_n^c, L_1)$ are discrete and ties are possible, we can still prove the upper bound. Consider rank of $R$ of $L_1$, i.e., put $\{L_1^c, \ldots, L_n^c, L_1\}$ in nondecreasing order, and let $R$ denote the position of $L_1$. When there are ties, position the tied values in a uniformly random way. By construction $R$ is uniformly distributed over $1, \ldots, (n+1)$ and

$$\mathbb{P}[L_1 \leq L_{(k^\star)}] \geq \mathbb{P}[R \geq k^\star] = \frac{k^\star}{1+n} \geq \alpha$$

$\square$

**Remark 3.** *For iid data and $m = 1$, the LAL-curve approaches the complementary cdf of $L_1$, i.e., $1 - F$. This facilitates the interpretation of the LAL-curve as a quantile point estimate. To see this, let $\widehat{F}_n^{-1}$ denote the empirical quantile function of $L_1$ based on the losses $L_1^c, \ldots, L_n^c$. Then the LAL of Thm. 2 can equivalently be defined as*

$$\ell_\alpha^\beta(\mathcal{D}) = \begin{cases} \widehat{F}_n^{-1}\left(\frac{n+1}{n}(1-\alpha)\right) & \text{if } \frac{n+1}{n}(1-\alpha) \in (0,1) \\ L_{(n+1)}^c & \text{else} \end{cases} \tag{9}$$

*If $L$ has a bounded and connected range, $\widehat{F}_n^{-1}$ converges uniformly to $F^{-1}$ (Bogoya et al., 2016), and $\ell_\alpha^\beta(\mathcal{D}) \to F^{-1}(1-\alpha)$. Plotting $\alpha$ on the vertical axis against $F^{-1}(1-\alpha)$ is identical to plotting the complementary cdf $1 - F(\ell)$ on the vertical axis against $\ell$, so in this case, the LAL-curve converges to the graph of the complementary cdf.*

### 3.3 LAL for large out-of-sample data sets

If the number of out-of-sample data points is very large, we may use a limit argument and let $m \to \infty$ in on Thm. 1.

**Corollary 1.** *Let $\text{BIN}^{-1}(\cdot; n, \beta)$ denote the quantile function of a binomial distribution with parameters $(n, \beta)$. For an infinite sequence of exchangeable losses $(L_1^c \ldots L_n^c, L_1, \ldots)$, let $\ell_\alpha^\beta(\mathcal{D}) = L_{(k^\star)}^c$ with*

$$k^\star = 1 + \text{BIN}^{-1}(1 - \alpha; n, \beta).$$

*This* LAL *satisfies a limit form of* (3)*:*

$$\lim_{m \to \infty} \mathbb{P}\left[L_{(\lceil m\beta \rceil)} > \ell_\alpha^\beta(\mathcal{D})\right] \leq \alpha \tag{10}$$

*Proof.* The result follows by expanding the binomial coefficients with Stirling's formula and taking the limit for $m \to \infty$. See App. A for details. $\square$

Starting from Cor. 1, we can find a different interpretation of the LAL when $m \to \infty$.

**Remark 4.** *Consider the case of iid data, where $L_i^c$ and $L_j$ have a common cdf $F$ for all $i, j$. Let $F^{-1}$ be the quantile function, and $\widehat{F}_m^{-1}$ be the empirical quantile function based on $\{L_1, \ldots, L_m\}$.*

*As $m \to \infty$, we have that $L_{(\lceil m\beta \rceil)} = \widehat{F}_m^{-1}(\beta) \to F^{-1}(\beta)$. Therefore,* (10) *simplifies to $\mathbb{P}\left[F^{-1}(\beta) > \ell_\alpha^\beta(\mathcal{D})\right] \leq \alpha$, which gives another interpretation of the LAL as the boundary of a confidence interval $C_\alpha^\beta(\mathcal{L}^c) := (-\infty, \ell_\alpha^\beta(\mathcal{D})]$. This interval satisfies $\mathbb{P}\left[F^{-1}(\beta) \notin C_\alpha^\beta(\mathcal{L}^c)\right] \leq \alpha$ and is thus a valid confidence interval for the $\beta$-quantile of $L_i$ for all $i$.*

*This intuition is also useful in the non-iid case as $m \to \infty$. The De Finetti theorem (Kingman, 1978) states that if $(L_1^c \ldots L_n^c, L_1, L_2 \ldots)$ forms an infinite sequence of exchangeable random variables, there is an auxiliary random variable $\zeta$ such that all the conditional variables $(L_i^c | \zeta)$ and $(L_j | \zeta)$ are iid with cdf $F_\zeta$. Therefore, the interpretation of the LAL as the boundary of a confidence interval is useful in the exchangeable-but-not-iid data setting, even if its interpretation must be handled with more caution.*

## 4 Experiments

This section presents examples of how the LAL can be applied to analyzing the out-of-sample loss of a model or family of models. Code to reproduce all experiments can be found at `https://github.com/el-hult/lal`.

The experiments are intended to be illustrative rather than exhaustive. They are limited in two respects. Firstly, to reduce runtime and simplify reproducibility of the experiments, we consider models $f$ that have relatively few parameters and are trained using small data sets. Since the theoretical guarantees hold for *any* fitted model, they also hold for large models and models trained on large data sets. Secondly, the experiments use small to moderate calibration set sizes $n$. As suggested in Remark 3, increasing $n$ improves the inference of loss distribution, but there is a practical limitation to be considered: When $n$ is large, it is advisable to transfer some data to the training set to improve the model performance, rather than improving our inferences about the model performance.

### 4.1 Study of asymptotics

This experiment illustrates the limit for different $m$ corresponding to Thm. 1, Cor. 1 and Thm. 2. The data set consist of California housing prices from the 1990 census (Kelley Pace & Barry, 1997), covering 20 640 housing blocks.

Each data point $z = (x, y)$ represents a single city block. The label $y \in \mathbb{R}$ is the median house value in the block, and the feature vector $x \in \mathbb{R}^8$ consists of continuous variables such as block coordinates, median house ages and tenant median income. The training data set $\mathcal{D}_0$ has $n_0 = 15\,000$ sampled without replacement. The calibration data set $\mathcal{D}$ has $n = 150$ and is sampled without replacement from the remaining data.

The model is a regression for the logarithm of the median house value, operating on standardized features and labels, and uses a random Fourier feature basis (Rahimi & Recht, 2008). Let $K$ random Fourier functions with bandwidth $b$ be stacked in a vector $\phi$. The model is $f(x) = \exp[\phi(x)^\mathsf{T} \hat{\theta}]$, where $\hat{\theta}$ is found by L2-regularized least squares regression. The hyperparameters (number of basis functions $K$, bandwidth $b$ and regularization strength $\lambda$) are tuned by five-fold cross-validation on the training data. The loss function is the absolute error expressed in dollars

$$\ell(x, y) = |y - f(x)|.$$

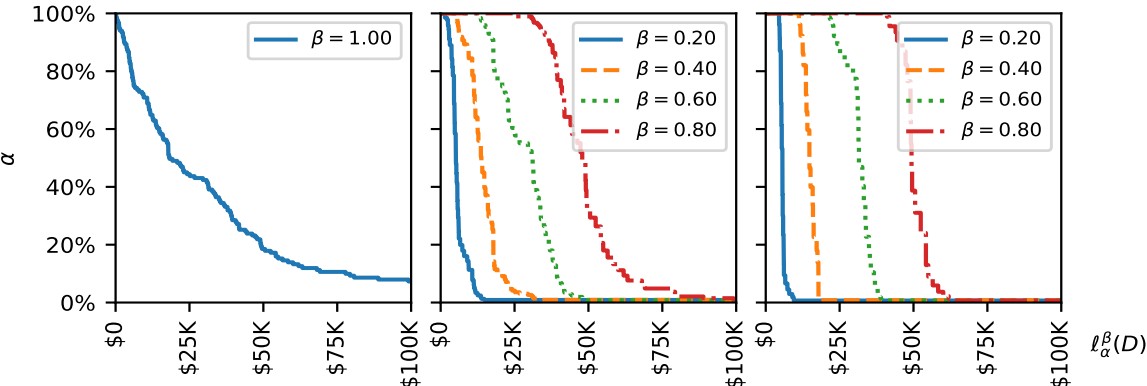

Figure 2: LAL-curves. From left to right, $m = 1$ computed with Thm. 2, $m = 30$ computed with Thm. 1 and $m = \infty$ computed with Cor. 1. The curves assure that a fraction $\beta$ of out-of-sample losses will not exceed $\ell_\alpha^\beta(\mathcal{D})$, with at least probability $1 - \alpha$. For example, we can see that among the $m = 30$ next samples, at least 80% of them will have prediction losses less than \$70,000, with a confidence of 90%.

We now turn to limiting $m$ out-of-sample losses, where $m \in \{1, 30, \infty\}$. LAL-curves were drawn with varying $\beta$-fractions, shown in Fig. 1 and Fig. 2. The empirical risk, defined as the average loss on the calibration data, is also calculated, and the calibration losses are presented as a histogram.

Averaging the losses over the calibration data gives an unbiased point estimate of the expected out-of-sample loss for $f$ but it, by itself, lacks statistical guarantees. While it estimates the mean out-of-sample prediction error to be around \$35 000, the LAL-curve for a single new prediction error ($m = 1$) in Fig. 2 shows that it may be close to \$80 000 (for $\alpha \approx 10\%$).

If we now consider a batch of $m = 30$ predictions, the LAL-curve for $m = 30$ in Fig. 2 informs us that $\beta = 80\%$ of them will have errors smaller than \$70 000 (for $\alpha \approx 10\%$). The number of data points blocks not analyzed are 5 490, so we may also consider the limiting case $m \to \infty$. The LAL curve now tells us that $\beta = 80\%$ of prediction errors will be smaller than \$60 000 (for $\alpha \approx 10\%$).

By comparing the LAL-curves in Fig. 2, we learn how the out-of-sample batch size $m$ affects the tail behavior of the LAL at a fixed $\beta$. The LAL-curve is slowly decaying for $m = 1$. For $m = 30$, the decay is faster. For $m = \infty$ the decay is even more abrupt. Some intuition can be gained for iid data. As $m$ increases, the variance of $L_{(\lceil m\beta \rceil)}$ decreases (it is asymptotically normal and $\sqrt{m}$-consistent, see e.g. (Vaart, 1998, Cor. 21.5)), and the bound in (3) can be made tighter.

### 4.2 Distribution shift analysis

This experiment illustrates the analysis of distribution shifts (Quiñonero-Candela et al., 2009), also known as concept drift, using the LAL-curve. The experiment also verifies Thm. 2 numerically. The availability of a calibration data $\mathcal{D}$ invites the notion of refitting or fine-tuning, to adapt to potential distribution shift (Lee et al., 2023). This is not always possible, e.g. when calibration data is limited (too small $n$). It may still be desirable to evaluate the model performance on out-of-distribution data. Since the LAL is valid for any $n$, it will always be an available diagnostic tool to analyze the extent to which distribution shift affects the model performance.

Lu et al. (2018) proposes a taxonomy for distribution shift detection methods. In this taxonomy, the LAL would qualify as an *error rate-based* method, as it is based on a loss function. Example methods in this group are the Drift Detection Method (DDM) (Gama et al., 2004) and Black Box Shift Estimation (BBSE) (Lipton et al., 2018) which detects distribution shift by monitoring when classifier performance changes with statistical significance. In contrast, the LAL does focus on distributions of out-of-sample losses, enabling analysis of *how/shift severity* for distribution shifts (again using the Lu et al. (2014) taxonomy). Other methods for *how*, focus on statistical distances between feature distributions (Rabanser et al., 2019). While

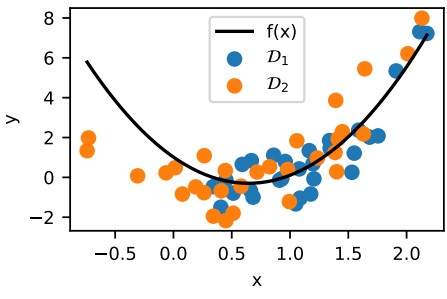

(a) Fitted model and calibration data

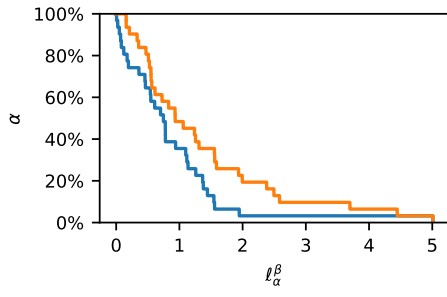

(b) LAL-curves, $m = 1$, $\beta = 1$

Figure 3: Quadratic regression model $f(x)$ evaluated using absolute prediction error loss function $\ell(x, y) = |y - f(x)|$. (a) A fitted model and two different calibration data sets $\mathcal{D}_1 \overset{iid}{\sim} p_1$ (identical to training data distribution) and $\mathcal{D}_2 \overset{iid}{\sim} p_2$ (shifted distribution). (b) LAL-curves under distributions $p_1$ and $p_2$. A single out-of-sample loss exceeds the limit $\ell_\alpha^\beta(\mathcal{D})$ by a probability of at most $\alpha$, as given by the curve. (See also Thm. 2).

such methods do not need labels, they do not measure shifts in the way that matters for the model, i.e., the loss.

Let $Z_i = (X_i, Y_i)$, with real valued $X_i$ and $Y_i$, and generate data according to

$$X_i \sim \mathcal{N}\left(\mu, \sigma^2\right) \tag{11}$$
$$Y_i | X_i = x \sim \mathcal{N}\left(x(x-1)(x+1), 1\right) \tag{12}$$

The training data set $\mathcal{D}_0$ was created with $n_0 = 100$, $\mu = 1$, $\sigma = 0.5$. A quadratic regression model was used, since it approximates the conditional expectation function well over the training data; $f(x) = \widehat{\theta}_0 + \widehat{\theta}_1 x + \widehat{\theta}_2 x^2$ was fitted to $\mathcal{D}_0$ via least-squares. See Fig. 3a. We analyze the case of out-of-sample batch size $m = 1$. The performance of the model is evaluated using the absolute error loss:

$$\ell(x, y) = |y - f(x)| \tag{13}$$

The out-of-sample losses are quantified for two different data distributions. In the first case, $\mu = 1$, $\sigma = 0.5$, so there is no distribution shift, and we use a calibration data set $\mathcal{D}_1$ for which $n = 30$. In the second case, $\mu = 0.75$, $\sigma = 0.75$, resulting in a shift in $X$ and we use a calibration data set $\mathcal{D}_2$ also having $n = 30$. The LAL-curves in both cases are presented in Fig. 3b, which reveal significantly larger out-of-sample losses for the distribution $p_2$ from which $\mathcal{D}_2$ was drawn. The 5%-tail losses are nearly twice as large for the shifted distribution.

Under the same experimental setup, we verified the tightness result of Thm. 2. Using 2 000 Monte Carlo runs, the empirical coverage was computed for different $\alpha$, and plotted in Fig. 4a. Moreover, Rem. 3 states the convergence of the LAL to the quantile function $F^{-1}(1 - \alpha)$ as $n$ increases. This is illustrated in Fig. 4b. The quantile function was numerically approximated using $10^5$ samples. The expected LAL is computed using 2 000 Monte Carlo runs with data sampled from $p_2$. We see that the LAL approaches the quantile function as $n$ increases.

### 4.3 Classification error analysis

This experiment shows that the proposed methodology can be applied to classification models as well. We will also consider how adversarial distribution shifts manifest in the LAL-curve. We use the Palmer Penguin data set, popularized by Horst et al. (2020). The 333 complete record data points are pairs $z_i = (x_i, y_i)$ of features $x_i \in \mathbb{R}^7$ and labels $y_i \in \{1, 2, 3\}$. The labels $y_i$ are categorical, encoding the penguin species. The features $x_i$ are vectors of (Island, Bill Length, Bill Depth, Flipper Length, Body Mass, Sex, Year), a mixture of categorical and integer-valued features.

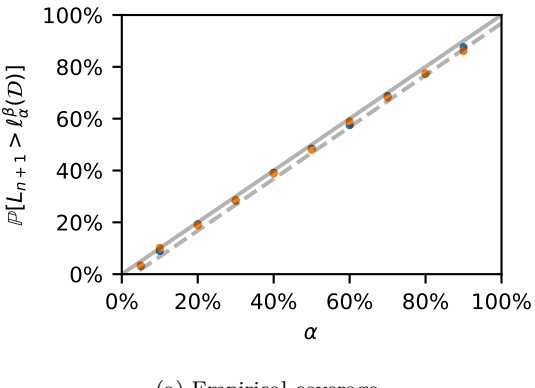
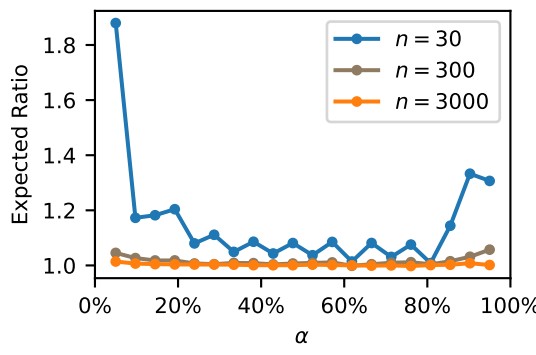

(a) Empirical coverage

(b) Convergence of the LAL to quantile function

Figure 4: The finite-sample guarantee of Thm. 2 is verified in (a), which illustrates both the upper bound (solid line) and lower bound (dashed line). (b) Illustration of the connection between LAL and the quantile function (Rem. 3). The expected value of the ratio $\ell_\alpha^\beta(\mathcal{D})/F^{-1}(1-\alpha)$ is evaluated for different values of $n$ with data drawn from the shifted distribution $p_2$. As $n$ increases, the expected ratio approaches 1, from above.

We use a training data set $\mathcal{D}_0$ of $n_0 = 150$, leaving 183 samples for calibration. Using $\mathcal{D}_0$, we fit $f(x)$ via multinomial logistic regression using L2-regularization and cross-validation. The model output is a three-dimensional vector $f(x) = [f_1(x), f_3(x), f_3(x)]^\mathsf{T}$ approximating the conditional probabilities, so that $f_i(x)$ approximates $\mathbb{P}[Y = i|X = x]$. The model is to be evaluated on calibration data using the misclassification probability loss

$$\ell(x, y) = 1 - f_y(x) \tag{14}$$

The out-of-sample batch size was set to $m = 1$.

Two different calibration data sets $\mathcal{D}_1$ and $\mathcal{D}_2$ of sample size $n = 50$ were constructed. We sampled from the 183 held out data points without replacement. For $\mathcal{D}_1$, the initial probability of selecting a sample was uniform over the data. For $\mathcal{D}_2$, the initial probability of selecting a sample was proportional to $\ell(x, y)$, effectively an adversarial reweighting of samples. The LAL-curves of Thm. 2 (set $m = 1$ and $\beta$ arbitrary) are presented in Fig. 5, showing that LAL-curves can be applied to classification models. The fact that one curve comes from an adversarial calibration sample is manifest by larger LAL for every confidence $\alpha$ compared to the non-adversarial calibration sample.

### 4.4 Regression error analysis

This experiment shows how alternative loss functions can be used to analyze the asymmetry of errors in regression problems. We use the UCI Airfoil data set (Dua & Graff, 2017). The task is to predict a label $y \in \mathbb{R}$ representing the sound measured in dB. The features $x \in \mathbb{R}^5$ Each data point is a feature-label pair $z_i = (x_i, y_i)$.

The calibration data $\mathcal{D}$ is constructed by weighted sampling of $n = 100$ samples. The probability to draw a data point $(x_i, y_i)$ is proportional to $\exp([1 \quad 0 \quad 0 \quad 0 \quad -1] \, x_i)$, making data points with high frequency and small displacement more likely to sample, similar to the distribution shift experiments in Tibshirani et al. (2020, sec. 2.2). The remaining 1 403 data points constitute the training data set $\mathcal{D}_0$.

The model $f(x)$ uses a spline basis that is fit using least-squares with L2-regularization and cross-validation. To study the asymmetry of prediction errors, we compute the LAL-curves for a subsequent experiment

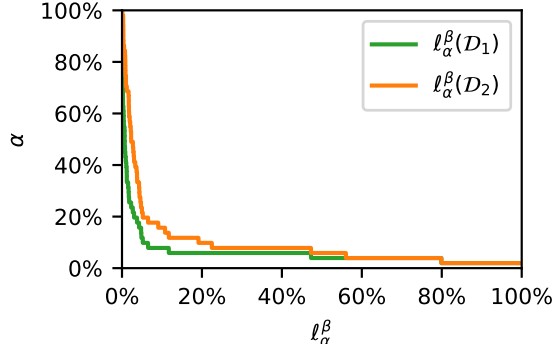

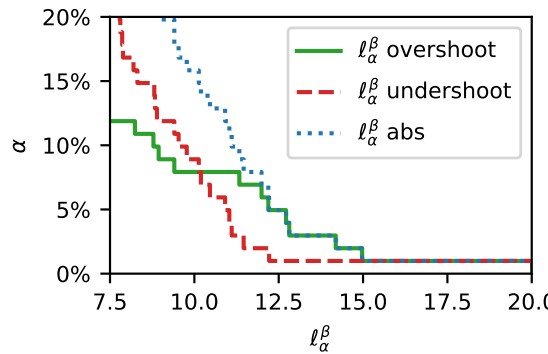

Figure 5: LAL-curves ($m = 1$) for the classification error analysis. The loss function indicates the certainty of misclassification – a loss of 80% means that the model assigned 80% probability to the wrong labels. Data set $\mathcal{D}_1$, is exchangeable with the training data. Data set $\mathcal{D}_2$ is adversarially sampled, presenting notably larger LAL.

Figure 6: LAL-curves ($m = 1$) for the regression error analysis. Using different loss functions, all in units of dB, gives deeper insight in the model fit. Looking at $\alpha$ around $2 - 8\%$, using loss for under- and overshoot, we see that the model is more likely to overshoot out-of-sample data than it is to undershoot.

($m = 1$) using three different loss functions,

$$\text{overshoot loss} \quad \ell(x, y) = \max(0, f(x) - y) \tag{15}$$

$$\text{undershoot loss} \quad \ell(x, y) = \max(0, y - f(x)) \tag{16}$$

$$\text{absolute loss} \quad \ell(x, y) = |y - f(x)| \tag{17}$$

which are all in units of dB to enable a physical interpretation. The results are shown in Fig. 6, where we observe that $f$ produces more severe overshoots than undershoots. The chosen loss functions (15-17) differs from the loss function used for model fitting (squared-error), illustrating the freedom that an analyst has to characterize the performance of $f$.

### 4.5 Model comparison

This experiment is concerned with model selection. The data used is the monthly number of earthquakes worldwide with magnitude $\geq 5$ between 2012 and 2022 (USGS, 2022). There are 120 data points $z_1, \ldots, z_{120}$, with $z_i \in \{0, 1, 2, \ldots\}$ These are randomly split into 100 and 20 data points, forming $\mathcal{D}_0$ and $\mathcal{D}$. We learn two models of the number of earthquakes per month $z$, $\mathbb{P}(Z = z)$, using the maximum likelihood method: a Poisson model $f^{\text{Poisson}}(z)$ and a Negative Binomial model $f^{\text{NegBin}}(z)$. The models are evaluated using the negative log-likelihood loss

$$\ell(z) = -\log f(z) \tag{18}$$

and Fig. 7 presents their respective LAL-curves for a subsequent earthquake, i.e., $m = 1$. From this result we can conclude that the simpler one-parameter Poisson model produces much larger out-of-sample losses than the two-parameter Negative Binomial model.

We compare the LAL analysis with a common model selection metric, the Akaike Information Criterion (AIC) (Ding et al., 2018). It evaluates the models as $\text{AIC}^{\text{Poisson}} = 1\,926.34$ and $\text{AIC}^{\text{NegBin}} = 1\,015.63$. One may also compare models by the average loss on the calibration data, yielding $5.69/4.78$ nats for the Poisson/NegBin. Both these model selection metrics favor the NegBin model, in agreement with the LAL-curve. Whereas the model selection metrics report a single number, the LAL-curve presents a more nuanced picture, showing the reduction in tail losses.

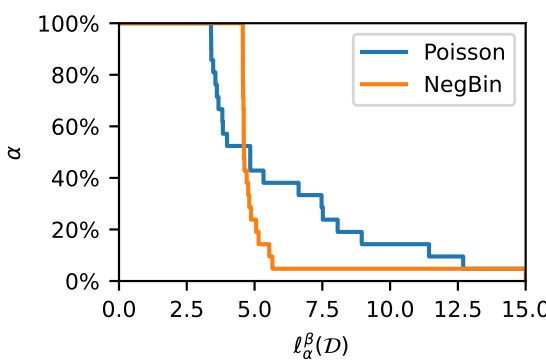

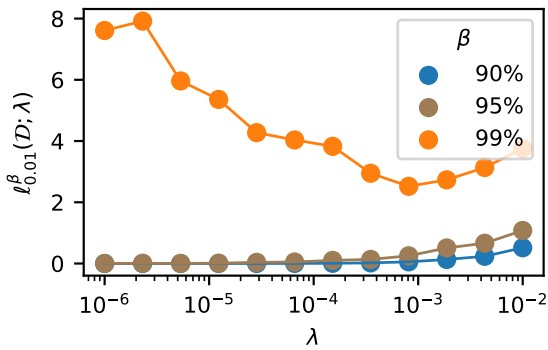

Figure 7: Comparison of models for earthquake statistics, using LAL curves and $m = 1$, $\beta = 1$. The loss function used is the negative log-likelihood of data under the fitted model. By considering how it handles the $\alpha = 20\%$ most difficult-to-fit data points, we see that the Negative Binomial model provides a better fit than the Poisson model.

Figure 8: Relation between a regularization parameter $\lambda$ in neural network model $f^\lambda(x)$, and the corresponding LAL for $m \to \infty$, varying $\beta$, and $(1 - \alpha)$ = 99% confidence. Appropriate regularization improves the performance in the tail of the loss distribution ($\beta = 99\%$), with an optimum around $\lambda \approx 10^{-3}$. The bulk of the losses ($\beta = 95\%$) are not reduced by regularization; their LAL increase with $\lambda$.

### 4.6 Hyper-parameter tuning

This experiment shows how the LAL-curve can be used for tuning a hyperparameter when training a neural network.

The data set is the MNIST handwritten digits (Bottou et al., 1994). The training data set $\mathcal{D}_0$ has $n_0 = 6 \cdot 10^4$ data points, and the calibration data set $\mathcal{D}$ has $n = 10^4$ data points. The features $x_i$ are images of handwritten digits, and the labels $y_i$ are integers 0-9. Data points are tuples $z_i = (x_i, y_i)$.

We construct a family of models $f^\lambda(x) = \text{NN}(x; \theta^\lambda)$. The function $\text{NN}(x; \theta^\lambda)$ consists of a dense feed-forward neural network with 3 hidden layers of 250 units each, parameters $\theta^\lambda$, ReLU activations and softmax output. The parameter $\theta^\lambda$ is learned by minimizing the cross entropy, using the Adam optimizer with an L2-regularization parameter $\lambda$ (aka. 'weight decay' in the deep learning literature). The optimization was run for 100 epochs, employing a batch size of 1024 and learning rate 0.01. The output of the network $f^\lambda(x)$ is a 10-dimensional vector, where the $i$th component approximates $\mathbb{P}[y = i | x]$.

We evaluate the model using the negative log-likelihood loss

$$\ell(x, y) = -\log f_y^\lambda(x) = -\log \left[ \text{NN}(x; \theta^\lambda) \right]_y \tag{19}$$

Since deep learning methods may be deployed without retraining over a large number of predictions, we consider the case $m \to \infty$ (Cor. 1) and study the LAL for a $\beta$-fraction of out-of-sample losses. The results in Fig. 8 show that small regularization initially reduces losses for outliers without increasing loss on nominal samples.

## 5 Discussion

This section elaborates on connections to related fields.

### 5.1 Model evaluation and selection techniques

Cross-validatory out-of-sample expected loss estimation is arguably the most common approach to model evaluation in machine learning (Arlot & Celisse, 2010; Stone, 1974; Bishop, 2006; Hastie et al., 2009). The

idea is that model performance is measured by expected loss on out-of-sample data (called the *risk*), and cross-validation estimates this quantity. In its most common form, a fraction of the training data is held out from model fitting. The average loss of the model is computed on the held out data, forming an estimate of the risk. Repeated splitting and refitting of the model (k-fold cross-validation) can be used to estimate the bias and variance of such estimates. Model evaluation with LAL shifts focus to evaluating model performance on probabalistic bounds on out-of-sample losses. This is significant when the out-of-sample loss distribution is multimodal, skewed or otherwise not well described by its mean and variance alone.

Evaluating models with respect to the risk may be done without cross-validation. Statistical learning theories, such as the VC-theory (Vapnik, 1991), provide asymptotic bounds on the risk under certain assumptions on the models and the data. Similarly, M-estimation (Vaart, 1998) provides another asymptotically valid method of fitting parametric models, quantifying the convergence of the average loss on training data to the risk. If the losses are bounded and the samples are iid or constructed as sampling without replacement, non-parametric results for inferring the risk appear to be promising (Waudby-Smith & Ramdas, 2020), improving on both the Hoeffding inequality and the Empirical Bernstein bounds (Audibert et al., 2009; Maurer & Pontil, 2009). For unbounded losses and non-iid exchangeable data, the inferential problem of estimating the risk remain open. LAL-curves provide a nonparemetric and nonasymptotic way to evaluate model performance by not using the risk as the quantity of comparison.

In this work, we have computed $\ell_\alpha^\beta(\mathcal{D})$ at given confidence levels $\alpha$. Conversely, one can interpret $\ell_\alpha^\beta(\mathcal{D})$ as the boundary value for rejection of the null hypothesis that model predictions are exchangeable, at level $\alpha$. Others have used hypothesis testing for model evaluation. Posterior predictive checks (Gelman et al., 2013; Rubin, 1984), and the data consistency criterion, (Lindholm et al., 2019) rely on exchangeability between observed data and data generated by the model to this end. Those methods are used to test whether a model is compatible with data, and reports a *p*-value for the test. A LAL-based analysis acknowledges that models are always misspecified and instead quantifies how well a model performs.

An analyst may wish to not only diagnose and compare models, but also decide which model in a set of candidates to use. This is called *model selection* (Ding et al., 2018). Two principal goals motivate the model selection algorithm: 1) model selection for inference, or 2) model selection for prediction. The first goal has been used to motivate well known information criteria such as the AIC (Stoica & Selen, 2004). Information critera has also been combined with hypothesis testing to assess if a certain model or model class can be said to be significantly closer to the true one than other candidates (Vuong, 1989). Regarding the second goal, Ding et al. (2018) posits that the 'best' model is the one with smallest expected out-of-sample loss, which may be estimated by cross-validation. By fixing $\alpha$ at any desired level, the LAL can be used to rank models in fashion similar to cross-validatory risk estimation. However, by employing the full LAL-curve as indicated in Sec. 4.5, a better informed decision can be taken by balancing typical versus outlier performance.

The Value-at-Risk (VaR) is a risk measurement used in the finance industry. For a random variable $L$ describing finanical loss, the value-at-risk at level $1 - \alpha$ can be defined as (Jorion, 2007, Eq 5.1)

$$\mathrm{VaR}_{1-\alpha}(L) = \inf\{x : \mathbb{P}[L > x] \le \alpha\}$$

Note that $x$ is not a random variable in the above expression. Comparing with (3) and fixing $m = 1$, we find that the VaR is the smallest LAL, conditional on the calibration data $\mathcal{D}$. Any VaR constitute a valid LAL for $m = 1$, but the converse is not true. Many VaR estimation methods rely on parametric assumptions or are only asymptotically valid. (Christoffersen, 2009; Alemany et al., 2013) The relaxed definition of LAL in contrast to VaR enables the distribution-free methodology presented in this article. For its use in e.g. financial risk measurements, VaR has been further developed into several variants such as Conditional VaR, Tilted VaR, Entropic VaR, and more. (Rockafellar & Uryasev, 2002; Li et al., 2021) This line of reasearch may be beneficial for the LAL as a diagnostic tool, and constitute a direction for further research, but some of the requirements on financial risk metrics may not necessarily carry over to all model performance metrics.

## 5.2 Non-parametric statistics and conformal prediction

For ease of exposition, the LAL has primarily been discussed as a point value $\ell_\alpha^\beta(\mathcal{D})$. One can also consider it as the boundary point of the interval $(-\infty, \ell_\alpha^\beta(\mathcal{D})]$. As such, it can be understood as a variant of a non-

parametric tolerance interval (Thm. 1), a prediction interval (Thm. 2) or a confidence interval (Rem. 4). See for instance Vardeman (1992) for a comparison between the different statistical intervals.

Fligner & Wolfe (1976) show how to construct non-asymptotic, non-parametric prediction intervals for quantiles of future data, and is a methodological precursor for the results in this paper.

Considering LAL a confidence interval of a quantile for iid data, there are other results with exact coverage (Zieliński & Zieliński, 2005), whereas the formula presented in this article is sometimes conservative. One could use that method with LAL to get exact guarantee. Such intervals are constructed via extra randomization and become harder to interpret. The LAL-curves in specific have no exact counterpart. We have therefore chosen to avoid this construction.

The theory of nonparametric prediction intervals also forms a foundation for conformal prediction. This field focuses on producing *prediction sets* for the output of any predictive model $f$, see e.g. Vovk et al. (2005) or Angelopoulos & Bates (2021) for introductions to the field. We will clarify the connection between conformal prediction and the LAL, in the case of split-conformal inference. The general case is essentially identical.

Consider a set of exchangeable random vectors $\{W_i\}_{i=1}^{M+1}$ taking values in $\mathcal{W}$. We wish to produce a prediction set $\mathcal{C}_\alpha\left(\{W_i\}_{i=1}^M\right)$ so that

$$\mathbb{P}[W_{M+1} \in \mathcal{C}_\alpha\left(\{W_i\}_{i=1}^M\right)] \geq 1 - \alpha$$

To this end, define a real-valued *nonconformity score* $A : W_i \mapsto A(W_i) = A_i$, with the semantic that a large value means $W_i$ does not conform to the general data set. Since the score is real valued, one can employ similar principles as in Thm. 2 to define a prediction interval $\mathcal{A}_\alpha$ such that

$$\mathbb{P}[A_{M+1} \in \mathcal{A}_\alpha\left(\{A_i\}_{i=1}^M\right)] \geq 1 - \alpha$$

By letting $\mathcal{C}_\alpha$ be the inverse image of $\mathcal{A}_\alpha\left(\{A_i\}\right)$ under $A$, i.e.,

$$\mathcal{C}_\alpha\left(\{W_i\}\right) = \left\{w | A(w) \in \mathcal{A}_\alpha\left(\{A_i\}_{i=1}^M\right)\right\},$$

we ensure the desired coverage. Conformal prediction methodology is thus largely centered on finding suitable nonconformity scores $A$ that are computationally tractable and handle various inference targets and data distributions (Angelopoulos & Bates, 2021).

More recently (Lei et al., 2018, Thm. 2.1), an upper bound on the coverage rate was derived,

$$1 - \alpha + \frac{1}{M+1} \geq \mathbb{P}[W_{M+1} \in \mathcal{C}_\alpha\left(\{W_i\}\right)] \geq 1 - \alpha,$$

that holds whenever the non-conformity scores are almost surely unique. The line of reasoning is similar but not identical to Thm. 2.

## 6 Conclusion

We have proposed the level-$\alpha$ loss (LAL) curve as a diagonistic tool for out-of-sample analysis of a model $f$. The method requires specifying a loss function of interest and the access to a calibration data set. In return it provides finite-sample guarantees about the probability of a batch of out-of-sample losses exceeding a certain threshold. The LAL is simple to compute and easy to interpret. A series of numerical experiments have been presented to show its usefulness in regression error analysis, distribution shift analysis, model selection and hyper-parameter tuning. We anticipate that there are many other areas of applications for this methodology.

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

## A   Details for Proof of Corollary 1

We must first show that

$$\lim_{m \to \infty} \frac{\binom{n-j+m-\lceil m\beta \rceil}{n-j}\binom{j+\lceil m\beta \rceil-1}{j}}{\binom{n+m}{m}} = \binom{n}{j}\beta^j(1-\beta)^{n-j}$$

By definition of binomial coefficient, this is

$$\lim_{m \to \infty} \frac{(n-j+m-\lceil m\beta \rceil)!(j+\lceil m\beta \rceil-1)!m!n!}{(n-j)!(m-\lceil m\beta \rceil)!j!(\lceil m\beta \rceil-1)!(n+m)!}$$

Rearrangement of factors gives

$$\binom{n}{j}\lim_{m \to \infty} \frac{(n-j+m-\lceil m\beta \rceil)!(j+\lceil m\beta \rceil-1)!m!}{(m-\lceil m\beta \rceil)!(\lceil m\beta \rceil-1)!(n+m)!}$$

So we must now show that

$$\lim_{m \to \infty} \underbrace{\frac{(n-j+m-\lceil m\beta \rceil)!(j+\lceil m\beta \rceil-1)!m!}{(m-\lceil m\beta \rceil)!(\lceil m\beta \rceil-1)!(n+m)!}}_{=:H(m)} = \beta^j(1-\beta)^{n-j}$$

By the upper and lower bounds in Stirling's formula (Robbins, 1955).

$$\sqrt{2\pi n}\left(\frac{n}{e}\right)^n e^{\frac{1}{12n+1}} < n! \le \sqrt{2\pi n}\left(\frac{n}{e}\right)^n e^{\frac{1}{12n}}$$

Let $\delta := \lceil m\beta \rceil - m\beta$. We introduce

$$
\begin{aligned}
h(m,Q) = &(2\pi(n-j+m(1-\beta)-\delta))^{1/2} && \left(\frac{n-j+m(1-\beta)-\delta}{e}\right)^{n-j+m(1-\beta)-\delta} && \exp\left[\frac{1}{12(n-j+m(1-\beta)-\delta)+Q}\right] \\
&\times (2\pi(j+m\beta+\delta-1))^{1/2} && \left(\frac{j+m\beta+\delta-1}{e}\right)^{j+m\beta+\delta-1} && \exp\left[\frac{1}{12(j+m\beta+\delta-1)+Q}\right] \\
&\times (2\pi m)^{1/2} && \left(\frac{m}{e}\right)^{m} && \exp\left[\frac{1}{12m+Q}\right] \\
&\times (2\pi(m(1-\beta)-\delta))^{-1/2} && \left(\frac{e}{m(1-\beta)-\delta}\right)^{m(1-\beta)-\delta} && \exp\left[\frac{-1}{12(m(1-\beta)-\delta)+(1-Q)}\right] \\
&\times (2\pi(m\beta-1+\delta))^{-1/2} && \left(\frac{e}{m\beta-1+\delta}\right)^{m\beta-1+\delta} && \exp\left[\frac{-1}{12(m\beta-1+\delta)+(1-Q)}\right] \\
&\times (2\pi(n+m))^{-1/2} && \left(\frac{e}{n+m}\right)^{n+m} && \exp\left[\frac{-1}{12(n+m)+(1-Q)}\right]
\end{aligned}
$$

allowing us to state succinctly

$$h(m,1) < H(m) \le h(m,0)$$

The proof is complete if we can show that $\lim_{m\to\infty} h(m,Q) = \beta^j(1-\beta)^{n-j}$. To see this we rearrange the factors. We will also use little-oh notation, i.e. $f(m) = o(g(m))$ iff $\lim_{m\to\infty} |f(m)|/g(m) = 0$. When taking limits, we use that $0 \le \delta < 1$ for all $m$.

$$
\begin{aligned}
h(m,Q) = &\underbrace{\left(\frac{2^3\pi^3}{2^3\pi^3}\frac{(n-j+m(1-\beta)-\delta)(j+m\beta+\delta-1)m}{(m(1-\beta)-\delta)(m\beta-1+\delta)(n+m)}\right)^{1/2}}_{=1+o(1)} \\
&\times \underbrace{e^{n-j+m(1-\beta)-\delta+j+m\beta+\delta-1+m-m(1-\beta)+\delta-m\beta+1-\delta-n-m}}_{=1} \\
&\times \underbrace{\left(\frac{n-j+m(1-\beta)-\delta}{m(1-\beta)-\delta}\right)^{m(1-\beta)-\delta}}_{=\exp(n-j)+o(1)} \underbrace{\left(\frac{j+m\beta+\delta-1}{m\beta+\delta-1}\right)^{m\beta+\delta-1}}_{=\exp(j)+o(1)} \underbrace{\left(\frac{m}{n+m}\right)^{m}}_{=\exp(-n)+o(1)} \\
&\times \underbrace{(n-j+m(1-\beta)-\delta)^{n-j}}_{=m^{n-j}(1-\beta)^{n-j}+o(m^{n-j})} \underbrace{(j+\beta m+\delta-1)^{j}}_{=m^j\beta^j+o(m^j)} \underbrace{(n+m)^{-n}}_{=m^{-n}+o(m^{-n})} \\
&\times \underbrace{\exp\left[(12(n-j+m(1-\beta)-\delta)+Q)^{-1}+(12(j+m\beta+\delta-1)+Q)^{-1}+(12m+Q)^{-1}\right]}_{=1+o(1)} \\
&\times \underbrace{\exp\left[-(12(m(1-\beta)-\delta)+(1-Q))^{-1}-(12(m\beta-1+\delta)+(1-Q))^{-1}-(12(n+m)+(1-Q))^{-1}\right]}_{=1+o(1)}
\end{aligned}
$$

By calculus of little-oh notation we find

$$h(m,Q) = \exp(n-j)\exp(j)\exp(-n)m^{n-j}m^j m^{-n}\beta^j(1-\beta)^{n-j} + o(1)$$

which in turn means

$$\lim_{m\to\infty} h(m,Q) = \beta^j(1-\beta)^{n-j}.$$

We have thus shown that the limit of (4) is

$$\lim_{m\to\infty} a(k) = \sum_{j=k}^{n} \binom{n}{j}\beta^j(1-\beta)^{n-j}.$$

By recognizing the binomial cumulative distribution function (and denoting it with the symbol $\text{BIN}(\cdot; n, \beta)$) we see

$$\lim_{m \to \infty} a(k) = 1 - \text{BIN}(k - 1; n, \beta).$$

Finally

$$k^\star = \min \left\{ k \in \{1, \ldots, n + 1\} \middle| \lim_{m \to \infty} a(k) \leq \alpha \right\} = 1 + \text{BIN}^{-1}(1 - \alpha; n, \beta)$$

## B   Extended Distribution Shift experiment

Continuing on the experiment on distribution shift, from Sec. 4.2. We keep $n = 30$, and also fix the random seed across runs to make the comparison clear. We generate calibration data sets $\mathcal{D}$ for three scenarios:

**Scenario 1, Increasing mean**
> The mean $\mu$ shift from 0.5 (no shift) to 3 (large shift), and the standard deviation $\sigma = 0.5$ (no shift).

**Scenario 2, Increasnig variance**
> The mean $\mu = 0.5$ (no shift), and the standard deviation shifts from $\sigma = 0.5$ (no shift) to 2 (large shift)

**Scenario 3, Decreasing variance**
> The mean $\mu = 0.5$ (no shift), and the standard deviation shifts from $\sigma = 0.5$ (no shift) to 0.05 (large shift)

These three series of calibration-sets have their LAL-curves plotted; see Fig. 9.

The results of Scenario 1 show that as the mean shift increase, model performance generally deteriorate. One detail should be noted for moderate mean-shifts. Some of the outlier data points with $x < 0.5$ contributed to a large LAL when there is no shift, but with increasing shift, these data points are moved towards 0.5, where model performance is generally better. This effect means that the LAL decreases somewhat, as can be observed for $\alpha \approx 10\%$. When the mean-shift increases, outliers with $x > 0.5$ produce larger losses, which is observed through the LAL-curve moving to the right. A large mean-shift moves most of the calibration set to $x$ where model performance is worse, as indicated by the whole LAL-curve shifting to the right, not only for selected $\alpha$.

The results of Scenario 2 show that the tail of the LAL-curve increase with feature variance, indicating that model performance on outliers get worse. Inliers are not as much affected.

The results of Scenario 3 show that the LAL-curve moves to the left with decreasing feature variance, indicating smaller losses. This means that the data concentrates in regions of feature space where the model performs well. We do observe some distribution shift, but this model still performs well on the data.

Taken together, this indicate that the LAL can indicate distribution shift, that it only detects shifts relevant for model performance, and that it indicates whether the shift incurs model degradation generally, or only on outliers.

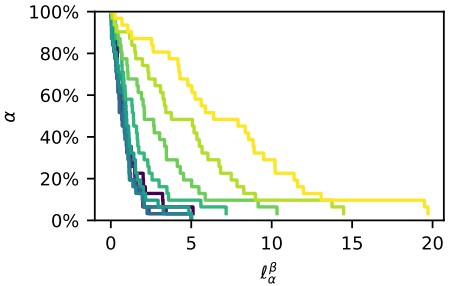

(a) LAL curves for distribution shift in Scenario 1, Increasing mean

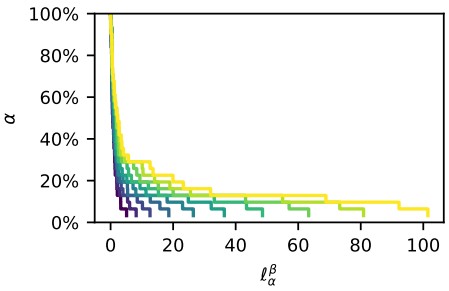

(b) LAL curves for distribution shift in Scenario 2, Increasnig variance

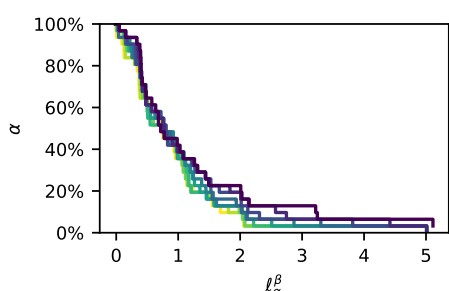

(c) LAL curves for distribution shift in Scenario 3, Decreasing variance

Figure 9: LAL-curves are drawn for various calibration sets $\mathcal{D}$ under distributions shift, as described in Appendix B. Dark blue line = no shift. Yellow line = large shift.

