# OpenReview forum: "Diagnostic Tool for Out-of-Sample Model Evaluation"
_TMLR — Accepted by TMLR_

### Review · Reviewer_DNsE · 2023-06-23

**Summary Of Contributions:**

This paper proposes a scalar metric called level-$\alpha$ limit for the $\beta$-fraction of $m$ losses, or LAL for short, where $m$ is the number of unknown out-of-sample input-output pairs. The first theorem shows how we can find this limit for a given fraction $\beta$ and probability $\alpha$ by using calibration data from the test distribution $p$. Then the paper shows another Theorem for the case with $m=1$.  The paper proves a corollary when $m\to\infty$. Various applications of LAL are shown in the experiments. Finally, the paper discusses related work such as model evaluation techniques, non-parametric statistics, and conformal prediction.

**Audience:**

Yes

**Broader Impact Concerns:**

There is no "Broader Impact Statement" section in this paper. I have no ethical concerns.

**Claims And Evidence:**

Yes

**Requested Changes:**

I would appreciate it if the authors can provide some discussions about the 3 aspects I raised in "weaknesses" in the previous section.

**Strengths And Weaknesses:**


Strengths
- The paper proposes a diagnostic tool, that can be used for various applications.
- It seems to be simple, easy to compute, and intuitive, such as $k^\star$ in Theorem 2.
- Figure 1 is motivating and provides an intuitive explanation to the problem and proposed method.
- The authors are willing to provide code to reproduce all experiments (while link was not provided in the manuscript to preserve anonymity).

Weaknesses
- The paper could benefit from more discussions about its advantages over existing metrics or ideas such as Value-at-Risk. For instance, Theorem 2's results seems to be similar to the procedure of deriving the VaR because we are choosing $k^\star$ to be $\lceil(n+1)(1-\alpha)\rceil$. It would be helpful if there are discussions about the differences and benefits over VaR.
- The practical benefit of utilizing LAL for distribution shift analysis appears somewhat unclear. Many transfer learning methods and domain adaptation methods study the situation where we have additional test samples during training. The paper's setup seems similar, because we have labelled samples from target distribution $p$. It would be natural to use this for training if we already know that there is a distribution shift. For identifying the potential shift, a more naive approach may be to just compare the two average test losses for $\mathcal{D}_1$ and $\mathcal{D}_2$. It would be helpful if the paper can have more motivating discussions for this application.
- Since the choice of $m$ appears somewhat arbitrary, the inclusion of guidelines for $m$ selection would improve the paper's practical value.

---

> ### Author Response · Authors · 2023-07-26
>
> We thank you for the review of our paper and your kind words. Below follows our replies regarding requested changes.
>
> > The paper could benefit from more discussions about its advantages over existing metrics or ideas such as Value-at-Risk [...] It would be helpful if there are discussions about the differences and benefits over VaR.
>
> The VaR does in a sense imply a LAL, providing a close connection between the concepts. We agree it would be beneficial to point out this connection explicitly, and explain how LAL and VaR are different in application. We will make this connection explicit in the discussion section of our paper.
>
> > The practical benefit of utilizing LAL for distribution shift analysis appears somewhat unclear. [...] It would be helpful if the paper can have more motivating discussions for this application.
>
> We acknowledge that the motivation for application in distribution shift quantification can be expanded. A typical case is small or moderate $n$, where the data from the new distribution is not enough to retrain or fine tune a model. This may occur e.g. in deployment of pretrained predictive models in the healthcare domain. Since LAL is valid for any sample size $n$, our methodology may still be utilized to characterize the performance of the model under this new distribution. Adding a discussion on these matters to the experiment on distribution shift (section 4.2) would be a service to the reader, and we are happy to do that.
>
> > Since the choice of $m$ appears somewhat arbitrary, the inclusion of guidelines for selection would improve the paper's practical value.
>
>  The choice of $m$ is entirely application specific, and the full range from $m=1$ and $m\rightarrow\infty$ may be useful. In the current writing, this is commented on in the various experiments. A collected discussion of the matter would improve the readability, and we will introduce that early in the article, where $m$ is defined.

---

### Review · Reviewer_dgnY · 2023-07-10

**Summary Of Contributions:**

This paper investigates a method to assess a model for comparing the models.  Unlike a traditional approach to evaluating a model, the authors try to characterize a batch of out-of-sample losses using a calibration data.  Eventually, finite-sample limits on the out-of-sample losses are studied and its statistical validity under general conditions is provided.  Finally, several numerical experiments, i.e., experiments on distribution shifts, model selection, and hyperparameter tuning, are demonstrated.

**Audience:**

Yes

**Broader Impact Concerns:**

I have no broader impact concerns.

**Claims And Evidence:**

No

**Requested Changes:**

1. Please specify why the proposed method is a "diagnostic" tool.  I think the term "diagnostic" is very ambiguous.
2. Comparisons to existing methods are missing.
3. The scale of experiments is small.  I do not think it is necessary for conducting additional large-scale experiments but can you say that the analysis presented in this paper still holds for the large-scale experiments?

**Strengths And Weaknesses:**

## Strengths

* It solves a simple but interesting method to evaluate a model.
* Various experimental circumstances are tested.

## Weaknesses

* No existing methods are compared.
* The scale of experiments is small.

---

> ### Author Response · Authors · 2023-07-26
>
> We thank you for a clear and brief review, being straight to the point. Below follows our replies to the requested changes.
>
> > Please specify why the proposed method is a "diagnostic" tool. I think the term "diagnostic" is very ambiguous.
>
>  The term "diagnostic" has a long tradition in statistics, mainly used about understanding the applicability of a certain model for a certain dataset. Common terms and phrases are "model diagnostics", "diagnostic checks", "regression diagnostics", "influence diagnostics" etc. Statistical and graphical tests are often employed. See e.g. *Semiparametric Regression'* by Ruppert (2003) https://doi.org/10.1017/CBO9780511755453 for multiple uses of the term.
>  It has also found its way into the ML community, in articles such as
>
>  - https://openreview.net/forum?id=l-kqvueSRp7 - Diagnostics for Deep Neural Networks with Automated Copy/Paste Attacks
>  - https://openreview.net/forum?id=losu6IAaPeB - DrML: Diagnosing and Rectifying Vision Models using Language
>
> We will point out this connection in a revised introduction.
>
>
> > Comparisons to existing methods are missing.
>
> Comparisons with other methods is highly application specific, so we reply on this per application.
>
> For the context of out-of-sample performance estimation, we have provided a comparison with cross validatory risk estimation, as presented in Figure 1.
>
> In the case of model comparison (experiment of section 4.5), well known methods with different motivations and philosophies include the Akaike Information Criterion and cross validatory average loss estimates. Comparing with these methods may clarify the concepts for the reader. We are happy to provide those metrics in an updated section 4.5.
>
> For distribution shift detection, there is an abundance of methods. Which one provides a suitable benchmark depends on application specific considerations, such as batch detection vs online detection, drift adjustment, feature-based vs outcome-based etc, as presented in the Lu 2018 paper referenced in section 4.2 (experiment on distribution shift). We are happy to position the LAL in such a taxonomy, relating it to known distribution shift methods.
>
> > The scale of experiments is small. I do not think it is necessary for conducting additional large-scale experiments but can you say that the analysis presented in this paper still holds for the large-scale experiments?
>
> We believe that the question of large scale machine learning have been addressed in this paper, in the following ways
>
>  - The scale of the *model* $f$ measured by the number of model parameters, only has an indirect influence. The whole paper is stated in terms of some fixed $f$ which may be a large scale machine learning model. Having a larger scale model increase the cost of time and compute to evaluate $f$. Only $m$ function evaluations are needed, and we typically consider $m$ relatively small. If $m$ is large, other methods can be used, e.g. employing the validation data in training, effectively reducing $m$.
>  - The scale of the *training data set* only comes into play for training the model $f$, assuming a parametric model. The LAL methodology analyzes the model after fitting, and thus becomes invariant to the training data size. In specific cases, such as k-nearest-neighbor or exact Gaussian Process, the training data size will influences model evaluation time -- see above point.
>  - The scale of the *test data set* is the parameter $n$. A large scale experiment, as measured by the asymptotic $n\to\infty$, is adressed contained in Remark 3 and Remark 4, and used in the MNIST experiment (section 4.6)
>  - The scale of the *calibration data set* is measured by $m$. The asymptotic for $m\to\infty$ is discussed in Corollary 1 and used in the MNIST experiment (section 4.6).
>
> Given the above motivation, we believe the MNIST experiment (section 4.6) suffices as a "large-scale" machine learning experiment.

---

### Review · Reviewer_bGoR · 2023-07-18

**Summary Of Contributions:**

This papers introduces an evaluation methodology for ML models based upon providing a bound the model's loss on new samples, which may be drawn from out-of-distribution data.

This "level-$\alpha$ limit" (LAL) provides an upper bound $\ell_\alpha^\beta(\mathcal D)$  such that with probability $1-\alpha$, a fraction $\beta$ of samples has a loss bounded by $\ell_\alpha^\beta(\mathcal D)$, where $\mathcal D$ is a calibration dataset drawn from the same distribution as the evaluation points (but not necessarily the training data distribution).

The authors provide a closed-form expression for a possible LAL in Theorem 1, and reinterpret this bound as a confidence interval boundary.

Finally, the authors evaluate their LAL across a variety of classification and regression tasks, with and without dataset shift.

**Audience:**

Yes

**Claims And Evidence:**

Yes

**Requested Changes:**

Clarifying the proof of thm.1, improving the positioning of the paper within existing literature.

**Strengths And Weaknesses:**

### Strengths
- The proposed bound is very intuitive, and fits well within concerns of reliability in ML.
- The proposed bound has a clear application to a variety of evaluation settings.
- Authors investigate a variety of settings (classification / regression, bounded / unbounded loss, in-distribution / out-of-distribution) experimentally.

### Weaknesses

A recurring question I have about this paper is how it fits within the context of the greater evaluation literature. In particular:
- For the proof of Thm 1, what components of the proof are required to deal with the non-iid data (departing from the Fligner & Wolfe (1976) work)?
- I'd be curious to see a detailed example comparing LAL to existing methods described in the literature (for example, expanding upon Figure 1 by adding to the cross-validation metric already provided). Even without an experimental analysis, discussing concretely where LAL addresses failure modes of existing methodologies (VC-theory, etc.) would help position this work with respect to the existing literature. For example, by assuming access to the calibration set drawn from the same distribution as the evaluation points, existing tools to deal with distribution shift (e..g, measures of discrepancy) are no longer necessary, and I suspect (?) that tighter bounds are possible.

On a more minor note,
- I found the proof of Theorem 1 to be very difficult to parse. Unless I a mistaken, $j$ is not defined. What are you choosing when you state that there are $\binom{n-j+m-i}{m-i}$ valid choices? It sounds like you are choosing an ordering, but earlier (before Thm 1) you state that the calibration and test sets are ordered.
- There are repeated results / statements in the experimental section. ("The label y is the median house value in the block. The training data set D0 has n0 = 15 000 sampled without replacement. The calibration data set D has n = 150 and is sampled without replacement from the remaining data", also the caption of Figure 2 is repeated in the main text).

### Related questions
- Many error bounds on models focus on worst-case analysis, which can be recovered by setting $\beta=1$ for LAL if I understand correctly. Do you recover similar results, or does the existence of the calibration set provide tighter bounds?
- On the experimental side, I really liked the OOD analysis. How does the bound evolve as you increase the type (mean/variance) and scale of shift?

---

> ### Author Response · Authors · 2023-07-26
>
>
> Thank you for your careful review. Below follows our replies regarding weaknesses and requested changes.
>
> > For the proof of Thm 1, what components of the proof are required to deal with the non-iid data (departing from the Fligner \& Wolfe (1976) work)?
>
> Whereas Fligner \& Wolfe use explicit formulas, working with integrals over the distribution of random variables and using that the joint distribution factorizes, our proof relies on a counting argument and the combinatorics of exchangeable random variables.
>
> > I'd be curious to see a detailed example comparing LAL to existing methods described in the literature (for example, expanding upon Figure 1 by adding to the cross-validation metric already provided). Even without an experimental analysis, discussing concretely where LAL addresses failure modes of existing methodologies (VC-theory, etc.) would help position this work with respect to the existing literature. For example, by assuming access to the calibration set drawn from the same distribution as the evaluation points, existing tools to deal with distribution shift (e.g. measures of discrepancy) are no longer necessary, and I suspect (?) that tighter bounds are possible.
>
> We will add a discussion on information criteria to section 5.2.
>
>
> > I found the proof of Theorem 1 to be very difficult to parse. Unless I a mistaken, $j$ is not defined. What are you choosing when you state that there are valid choices? It sounds like you are choosing an ordering, but earlier (before Thm 1) you state that the calibration and test sets are ordered.
>
> A mistake in the editing of the proof made $j$ undefined. Equation 5 should read
>
> $ \mathbb{P}[L_{(j)}^c \leq L_{(i)} \leq L_{(j+1)}^c] = \frac{\binom{n-j+m-i}{n-j}\binom{j+1-1}{j}}{\binom{n+m}{m}} $
>
> and $j \in \\{1,\dots n\\}$ is a free variable in the left-hand side.
> We thank you for pointing this out and will correct the presentation of the proof. We will also edit the proof, trying to make it simpler to read and follow.
>
> Regarding the ordering, the ordinal statistics $L_{(i)}$ are ordered values of the random variables $L_i$, as is standard in statistics. The "choices" refer to the various ways such an ordering can be constructed. It is not a choice we make as analysts in machine learning, but simply a phrasing to convey a combinatorical argument. We will rephrase this to reduce ambiguity.
>
> > There are repeated results / statements in the experimental section. ("The label y is the median house value in the block. The training data set D0 has n0 = 15 000 sampled without replacement. The calibration data set D has n = 150 and is sampled without replacement from the remaining data", also the caption of Figure 2 is repeated in the main text).
>
> We will remove the duplication of text in section 4.1. Thank you for pointing this out. Some repetition of information in the caption for Figure 2 and main text (section 4.1) is intentional, and it aims to improve the reading experience both for those readers who just skims the figures, and those reading the details. As the repetition is not verbatim, we think this is acceptable.
>
>
> > Many error bounds on models focus on worst-case analysis, which can be recovered by setting $\beta=1$ for LAL if I understand correctly. Do you recover similar results, or does the existence of the calibration set provide tighter bounds?
>
> Indeed, setting $\beta=1$ is a kind of worst-case analysis - you obtain the LAL associated with the worst loss among $m$ out-of-sample data points.
>
> Unfortunately, we are not certain about which bounds you refer to as comparison. One variant of "worst case" analysis is in distributionally robust opmimization (DRO). That worst-case analysis is however not really comparable to the sense here. The "worst case" in DRO refers to adversarial selection of a distribution to sample, whereas the LAL with $\beta=1$ only produces a worst-case analysis is terms of worst-outcome-from-a-sample.
>
> > On the experimental side, I really liked the OOD analysis. How does the bound evolve as you increase the type (mean/variance) and scale of shift?
>
> We will add such a numerical anlysis in the appendix, as it may be illustrative.
>
> > Requested change: Clarifying the proof of thm.1
>
> We are happy to clarify the proof of Thm. 1, as discussed above.
>
> > Requested change: Improving the positioning of the paper within existing literature
>
> Based of yours and the other reviwers comments, we understand that there is need to expand this. We will add a connection to model selection in the discussion (section 5) and expand the link to distribution shift analysis in section 4.2.

---

### Decision · Action_Editors · 2023-09-04

**Recommendation:** Accept with minor revision

**Comment:**

Most of the reviewers and myself found the paper well written. The authors motivate well the problem they want to tackle and nicely introduce the notions and results they need to develop their methodology.

One of the reviewers found that the experiments were limited, a viewpoint with which I agree. However, I believe that as an initial introduction, the experiments conducted by the authors suffice as a proof of concept to justify their new tools. This is why I have decided to accept the paper.

Please take into consideration reviewers' comments before submitting the final version of your work.
In particular, I found the proof of Theorem 1 a little hard to follow. Could the authors formalize their arguments a little more to improve readability?
Finally, the authors should acknowledge that their experiments are small-scale and discuss how larger-scale experiments could be conducted."

**Audience:**

Model diagnostics using out-of-sample analysis is of interest in many applications and the publication then appears relevant for the TMLR audience.

**Claims And Evidence:**

This papers introduces a new quantity for performing diagnostics on trained ML models based on new samples, which may be drawn from out-of-distribution data.

This new quantity $\ell_{\alpha}^\beta (\mathcal{D})$, based on a calibration set $\mathcal{D}$, is defined for a level $\alpha \in (0,1)$, a fraction $\beta  \in (0,1)$ and $m$ out-of-sample data points, as any value such that the probability that at least a fraction $\beta$ of the out-of-samples losses are smaller than $\ell_{\alpha}^\beta (\mathcal{D})$, is larger than $1-\alpha$.

The first result presented by the authors is the derivation of explicit expressions for $\ell_{\alpha}^\beta (\mathcal{D})$, with a straightforward one for $m=1$ and a limiting one as $m\to \infty$.

This new metric is relatively simple, and the significance of this new criterion is illustrated across a wide range of ML applications, including distribution shift analysis, classification error analysis, regression error, model comparison, and hyper-parameter tuning.